# Effects of Microbeam Irradiation on Rodent Esophageal Smooth Muscle Contraction

**DOI:** 10.3390/cells12010176

**Published:** 2022-12-31

**Authors:** Bernd Frerker, Stefan Fiedler, Timo Kirschstein, Falko Lange, Katrin Porath, Tina Sellmann, Leonie Kutzner, Fabian Wilde, Julian Moosmann, Rüdiger Köhling, Guido Hildebrandt, Elisabeth Schültke

**Affiliations:** 1Department of Radiation Oncology, Rostock University Medical Center, 18059 Rostock, Germany; 2Oscar Langendorff Institute of Physiology, Rostock University Medical Center, 18057 Rostock, Germany; 3European Molecular Biology Laboratory, 22607 Hamburg, Germany; 4Center for Transdisciplinary Neurosciences Rostock, Rostock University Medical Center, 18147 Rostock, Germany; 5Helmholtz-Zentrum Hereon, X-ray Imaging with Synchrotron Radiation, Hereon Outstation, 21502 Geesthacht, Germany

**Keywords:** high-dose-rate radiotherapy, microbeam irradiation, broadbeam irradiation, organ bath, esophageal smooth muscle, carbachol-induced contraction, esophageal function and motility

## Abstract

Background: High-dose-rate radiotherapy has shown promising results with respect to normal tissue preservation. We developed an ex vivo model to study the physiological effects of experimental radiotherapy in the rodent esophageal smooth muscle. Methods: We assessed the physiological parameters of the esophageal function in ex vivo preparations of the proximal, middle, and distal segments in the organ bath. High-dose-rate synchrotron irradiation was conducted using both the microbeam irradiation (MBI) technique with peak doses greater than 200 Gy and broadbeam irradiation (BBI) with doses ranging between 3.5–4 Gy. Results: Neither MBI nor BBI affected the function of the contractile apparatus. While peak latency and maximal force change were not affected in the BBI group, and no changes were seen in the proximal esophagus segments after MBI, a significant increase in peak latency and a decrease in maximal force change was observed in the middle and distal esophageal segments. Conclusion: No severe changes in physiological parameters of esophageal contraction were determined after high-dose-rate radiotherapy in our model, but our results indicate a delayed esophageal function. From the clinical perspective, the observed increase in peak latency and decreased maximal force change may indicate delayed esophageal transit.

## 1. Introduction

Radiation-induced esophagitis is one of the most common [1] and dose-limiting [2] acute toxicities in treating various thoracic tumors. After reaching a cumulative dose of 20 to 30 Gy of conventional fractionated radiation therapy, the affected patients may suffer from dysphagia or odynophagia [3], often necessitating symptomatic therapy [4]. In most cases, these symptoms are associated with esophagitis [5], which is due to radiation-induced mucosal damage [6]. Nevertheless, morphological mucosal changes are not always present, and in some patients, the clinical symptoms do not necessarily correlate with the endoscopic findings [5]. While some patients complain of only low-grade dysphagia despite endoscopically more severe esophagitis [6], other patients with subjectively higher-grade dysphagia had only marginal or even absent endoscopic findings [7]. Alternatively, there is evidence of radiation-induced motility disorder (RIMD). In the esophagus, RIMD is usually described as a late sequel [8,9,10,11] due to damaged esophageal muscle layers [8,10] or nerves [8], mainly associated with fibrosis or stenosis [8,9,10], but has also been described as acute toxicity [12,13,14]. In association with several clinical trials, it was confirmed that esophageal transit is acutely impaired by irradiation [15,16,17]. However, these findings remain controversial [16,17], as other trials found no effect [7] following irradiation. The QUANTEC database [18] discussed acute esophagitis in detail, whereas abnormal esophageal motility was mentioned only once. The underlying mechanisms are insufficiently characterized and, consequently, difficult to treat.

Over the last decade, high-dose-rate irradiation has increasingly come into focus for superior preservation of normal tissue [19]. Microbeam irradiation (MBI) is an experimental irradiation technique characterized by a high dose rate and spatial dose fractionation of synchrotron-generated X-ray beams in the micrometer range [19]. A multislit collimator (MSC) is inserted into the X-ray beam, producing an array of quasi-parallel microbeams. This results in an inhomogeneous dose distribution in the irradiated target, with a repetitive sequence of high (peak) dose and low (valley) dose zones [20]. The width of individual microbeams is typically in the range of 20–100 µm with a center-to-center spacing of several hundred micrometers. With its high dose rates, MBI takes advantage of the FLASH effect, described as tissue-preserving at dose rates of ≥40 Gy/s [21,22]. It has shown effective tumor control in small animal models and excellent normal tissue tolerance in the brain [23,24,25,26,27,28,29]. In a recent study, a spontaneous canine brain tumor was successfully treated with MBI [30]. While the initial focus of MBI development was in the brain, more recent studies have also shown good normal tissue tolerance in the lung [31] and efficiency in treating lung tumors in a small animal model [32]. In an ex vivo study, it was shown that, even with peak doses up to 400 Gy, MBI could be conducted without severe acute effects on cardiac function [33,34]. In addition to the heart, the esophagus is also an organ at risk (OAR) in thoracic irradiation. To our knowledge, no previously published study investigated the effect of high-dose-rate irradiation on the esophagus. Therefore, we designed a pilot study to develop an ex vivo model system suitable for assessing the acute effects of irradiation on esophageal function. Our major finding is a delayed esophageal contraction without loss of contraction strength following MBI.

## 2. Materials and Methods

### 2.1. Preparation of Isolated Rat Esophageal Segments for Isometric Contraction Measurement in the Organ Bath

All experiments were conducted in accordance with the Guide for the Care and Use of Laboratory Animals. Sufficient water and food were available. In the current study, an acute ex vivo model was used. Before any experiment or procedure, Wistar rats aged 8–12 weeks were anesthetized and decapitated under deep anesthesia. Deep anesthesia was proven by the absence of pain reflex. After the post-mortem esophagectomy, the esophagus was submerged into a HEPES-buffered storage solution (in mmol/L: 120 NaCl, 4.5 KCl, 26 NaHCO_3_, 1.2 NaH_2_PO_4_, 1.6 CaCl_2_, 1.0 MgSO_4_, 0.025 Na_2_-EDTA, 5.5 glucose, 5.0 HEPES, pH = 7.4) for the preparation, and the esophagus was cut into three sections representing the proximal, middle, and distal part of the esophagus. For isometric contraction measurements, as described before [35,36], thin nylon threads (Gütermann Toldi) were sutured to either end of the segments to enable longitudinal fixation in an organ bath (Panlab ML0146/C, ADInstruments, Oxford, UK). The organ bath was filled with a buffer solution (in mmol/l: 120 NaCl, 4.7 KCl, 2.5 CaCl_2_, 1.2 MgCl_2_, 30 NaHCO_3_, 0.5 Na_2_-EDTA, 5.5 Glucose, 2.0 sodium-pyruvate, pH = 7.4, osmolarity 295–300 mosmol/L) and continuously gassed with carbogen (95% O_2_ and 5% CO_2_, AirLiquide, Lutherstadt-Wittenberg, Germany). The temperature in the organ bath was kept at 37 °C. Isometric contraction was measured with a force transducer (MLT0201, ADInstruments, Oxford, UK) and recorded with a bridge amplifier (ML224, ADInstruments) connected to an analog-digital-converter (Powerlab 4/30, ADInstruments) and analyzed by the LabChart 7 Software (ADInstruments).

### 2.2. Time Course of the Experiments

The time course of the experiments and a representative measurement of the contraction force are shown in Figure 1. After fixation in the organ bath, the initial mean tension of the segments (4.56 mN ± 0.16 mN, *n* = 79) was adjusted, and recordings were registered for 30 min to establish stable baseline conditions. Then, this baseline tone was recorded for 15 min (Figure 1A). Carbachol (carbamoylcholine chloride, CCH, Tocris bioscience, Bristol, UK), a structural non-hydrolyzable analog of the neurotransmitter acetylcholine, was used to induce the isometric contraction. After adding 100 µL of the CCH stock solution (2.5 mM CCH) to yield a final CCH concentration of 10 µM in the organ bath, the isometric contraction was recorded for 15 min (Figure 1B). CCH was washed-out, the specimens were allowed to relax, and the baseline tone was reached after an additional duration of 30 ± 10 min. The segments were removed from the organ bath with their force transducers to maintain the tension and then positioned on the irradiation table and irradiated. The segments were dipped into the buffer solution just before irradiation to keep them as humid as possible. After irradiation, the force transducers and the segments were returned to the organ bath. The time interval between the removal of segments and replacing them in the organ bath was 8 ± 3 min. The segments recovered for 15 min. Finally, 100 µL of 2.5 mM CCH was added again, and isometric contraction was registered for another 15 min. To compare the effects of MBI with its high peak doses to the effects caused by a homogeneous valley dose, we also performed a high-dose-rate BBI study with a dose approximately corresponding to the valley dose of the MBI study (first control group, Table 1). For technical reasons, during the irradiation process, the segments were not submerged in the buffer solution. Thus, to control for this situation, we also performed sham irradiation as a second control group (Table 1). These control segments were not irradiated, but the buffer solution was removed for 8 ± 3 min, equivalent to the duration of the irradiation procedure. At the end of the experiments, all segments were fixed in 3.7% PFA-solution for immunohistochemistry.

### 2.3. Calculation of Parameters for Characterization of Isometric Contraction

The force-time curves were used to calculate several parameters (Figure 1C). In addition to the segment length, five parameters were calculated. To describe the function of the contractile apparatus, we calculated the baseline tone, the maximal contraction strength, and the peak amplitude. The peak latency and the maximal force change were calculated to evaluate the signal transduction process. The baseline tone was the mean contraction strength during the last 10 s before adding CCH, and the peak amplitude was the difference between maximal contraction strength and baseline tone. Peak latency was the interval between adding CCH and the maximal contraction strength. The force-time function was derived from the recorded force-time curves, and the inflection point (dF/dt) was calculated.

### 2.4. Irradiation Protocol

The experiments were conducted at the beamline P05 of the PETRA III synchrotron-radiation source operated by HEREON on the DESY campus in Hamburg, Germany [37]. The setup was designed as a full-field imaging beamline with a tunable monochromatic energy spectrum between 5 keV and 50 keV. The irradiation procedure was conducted in the second experimental hutch, dedicated to microtomography. Due to the large distance of 85.9 m from the undulator source, a beam width of up to 7 mm horizontally was obtained. To operate with an optimized photon flux, the experiment was carried out with a double multilayer monochromator at an energy of 30 keV with an energy bandwidth of approximately 1%. This can deliver a photon flux in the order of 10^13^ ph/s. Although MBI studies have been conducted in the past exclusively at white beam beamlines, mostly on wiggler sources, from a physics perspective, there are certain advantages to using a monochromatic beam. A beam produced by an undulator offers an intrinsically better horizontal collimation (at P05 28 µrad rms), improving the homogeneity of off-axis microbeams. Due to the monochromatizating, no beam hardening occurs. With the photon energy fluence, in general, well determined at a monochromatic synchrotron beamline, the absorbed dose rate to a medium can be readily calculated using the mass energy absorption coefficient [38]. The drawback of an undulator beamline is, however, that the dose rate is at least one order of magnitude lower compared to the one available at a white beam wiggler source. For ex vivo experiments, such as the one developed in this study, given that the motility of the esophagus is only minimal, the requirements of an extremely high dose rate are not as strict as they would be for in vivo experiments where much physiologic movement due to breathing and cardiac activity can be expected. In the case of a physiologic movement during irradiation, a lower dose rate and a subsequently longer exposure time can cause a smearing of the microbeam edges, which in turn will result in a decrease of the peak-to-valley dose ratio (PVDR) and thus impairment of normal tissue tolerance. There is evidence in experimental radiotherapy for a positive correlation between dose rate and normal tissue protection [21]. For this experiment, the beam size was adjusted to 4.85 × 3.8 mm^2^ (horizontal × vertical), optimizing the homogeneity of the intensity distribution. The dose rate of the broadbeam field was determined with a small field, soft X-ray ion chamber for clinical use (PTW TM34103W) and cross-checked with a Si photodiode (Canberra PIPS detector calibrated at PTB Berlin, the German national institute for metrology standards). To determine the dose in the microbeams, a method providing a high spatial resolution was necessary. To achieve this, Gafchromic™ film (HD-V2, EBT3, Ashland, Bridgewater, MA, USA) was used after cross-calibration in the broadbeam field. A dose rate of 81 Gy/s in the broadbeam field was measured. All dosimetric values presented here were determined at the sample entrance. The broadbeam field was split by an MSC (UNT, Morbier, France) into an array of vertical quasi-parallel microbeams with an individual width of 50 µm, spaced at a center-to-center distance of 400 µm. Using the available horizontal beam width, up to 12 microbeams could be obtained. Thus, the target zone was covered with a grid of high (peak) dose zones and low (valley) dose zones. The microbeam array could be visualized with a CMOS camera used for microtomography (Ximea CB500MG, 7920 × 6004 pixels with a pixel size of 0.9 microns at the lowest magnification). The esophagus segments were irradiated either in MBI mode (*n* = 39, group I in Table 1) or in BBI mode (seamless irradiation, *n* = 20, group II). Some segments served as non-irradiated controls (*n* = 20, group III). The irradiation dose was administered in one single fraction in both MBI and BBI modes. During the irradiation, the samples were translated vertically through the beam. By varying the speed or the vertical beam height, the dose deposited in the sample was altered by two orders of magnitude. For the high peak dose MBI, a speed of 0.77 mm/s was chosen. For the low-dose BBI, the speed was raised to 2.18 mm/s while reducing the beam height to 0.15 mm. For dosimetry and verification of the correct positioning, a Gafchromic^TM^ film was placed behind the irradiated specimens.

### 2.5. Immunofluorescence Labelling

Four to eight hours after irradiation, the fixed specimens were cryoprotected in 30% sucrose solution for 24 h and frozen in 2-methylbutane. The horizontal sections (20–30 µm, CM3050S, Leica Biosystems Nussloch GmbH, Nußloch, Germany) were treated for 10 min at 95 °C with citrate buffer pH 6.0 (antigen retrieval), then washed three times and permeabilized with TritonX-100-Solution (Sigma Aldrich, St. Louis, MO, USA). After blocking with 10% normal-goat-serum (NGS, Sigma Aldrich, St. Louis, MO, USA) for 1 h, the specimens were treated with anti-gamma H2AX antibody (1:200; ab2893, Abcam, Cambridge, UK) or anti-desmin antibody (Sigma Aldrich, St. Louis, MO, USA) overnight at 4 °C. After several washes the samples were incubated for 2 h at 37 °C with the secondary goat anti-rabbit IgG Cy3 antibody (1:200; Invitrogen A10520). The slices were washed three times, covered with ProLong Gold antifade mountant with DAPI (Invitrogen, Waltham, MA, USA), and viewed and quantified with LAS software on a Leica DMI6000 microscope.

### 2.6. Statistical Analysis

SigmaPlot (Systat Software Version 13) was used for the statistical analysis. The test of univariate normality was the Shapiro-Wilk test. To compare the data before and after irradiation, a parametric paired *t*-test and a non-parametric Wilcoxon test were performed. For comparison between the three groups, a non-parametric Kruskal-Wallis test followed by a post hoc test (one-way ANOVA on ranks) was used. In some cases, we also used a parametric two-sample student’s *t*-test, a non-parametric Mann-Whitney U-test, and a parametric one-way ANOVA. The level of significance was set to 0.05. The data are presented as mean ± standard error of the mean (SEM).

## 3. Results

### 3.1. Concentration-Response Relationship for Carbachol

The first experiment was to investigate the concentration dependence of CCH-induced contraction of the esophageal smooth muscle (Appendix A). Before adding CCH, the baseline tone among the segments was not statistically different (mean baseline tone of all segments 2.99 ± 0.52 mN, one-way ANOVA, *p* = 0.252, grey box in Appendix A). In the presence of 0.01 µM CCH, no contraction was registered. The recorded force remained within the range of the baseline tone (student’s *t*-test, *p* = 0.739). With increasing concentrations of CCH, discernible isometric contractions were obtained with increasing amplitudes until all muscle cells were contracted and a plateau was reached. Flattening of the dose-response curve was presumed in the presence of 10 or 100 µM CCH indicating saturation (11.32 ± 1.33 mN, or, respectively, 12.20 ± 1.75 mN). There was no statistical difference between these two contractions (student’s *t*-test, *p* = 0.715). However, the relaxation time after washing out CCH was longer for the higher concentration of CCH. Therefore, we conducted the rest of the study using 10 µM CCH concentration in the organ bath (dashed rectangle, Appendix A).

### 3.2. The Three Experimental Groups Were Homogeneously Randomised

To ensure that the segments were evenly distributed throughout the groups, we analyzed the CCH-induced contraction in all segments before irradiation, including the mean segment length, mean baseline tone, mean maximal contraction strength (peak), mean peak amplitude, mean peak latency, and mean maximal force change. We found no significant differences (Kruskal-Wallis ANOVA) on the rank test (Table 2) between the experimental groups (Table 1).

### 3.3. MBI Significantly Increased Peak Latency and Decreased Maximal Force Change

We next analyzed the peak latency and the maximal force change (Figure 2A,B) before and after irradiation. In the BBI group, we found no significant difference (paired *t*-test resp. Wilcoxon test, see Figure 2A,B). However, in the MBI group, the peak latency significantly increased (260 ± 31 s to 335 ± 33 s, Wilcoxon test, *p* < 0.001), and the maximal force change significantly decreased (0.12 ± 0.02 mN/s to 0.09 ± 0.01 mN/s, Wilcoxon test, *p* < 0.001). This may indicate an impairment of the signal transduction cascade for contraction in muscle cells after MBI. Furthermore, the contraction in the SHAM group was not altered, suggesting that the irradiation procedure did not affect the CCH-induced contraction.

### 3.4. Subgroup Analysis of the Peak Latency and Maximal Force Change

Since the peak latency significantly increased and the maximal force change significantly decreased following MBI, we did a subgroup analysis in this group (Figure 2C,D). Only peak latency for the distal segment (Wilcoxon test, *p* = 0.027) and maximal force change for the middle (Wilcoxon test, *p* = 0.005) and distal (Wilcoxon test, *p* = 0.021) segment remained significantly different. In the proximal segment, the peak latency (Wilcoxon test, *p* = 0.091) and the maximal force change (paired *t*-test, *p* = 0.106) were not altered.

### 3.5. MBI Did Not Affect Baseline Tone, Maximal Contraction Strength, and Peak Amplitude

We also compared the baseline tone, maximal contraction strength, and peak amplitude to characterize the contractile apparatus (Figure 3A–C). We found no significant differences within the MBI- and the BBI-group (paired *t*-test resp. Wilcoxon-test if normality test failed, *p*-values see Figure 3). This indicates that MBI and BBI did not affect the contractile apparatus of esophageal smooth muscle. Again, there was no statistical difference in the SHAM group.

### 3.6. Dosimetric Characteristics

Based on the HD-V2 Gafchromic™ film measurements (Figure 4A), the peak dose was between 225 ± 15 Gy. The microbeam profile (Figure 4B) shows that the higher doses were delivered in the more central the microbeams, while the peak doses were lower in the peripheral microbeams. Since the incident beam width at the sample position was only 5.6 mm, only the centrally located 12 slits of the MSC were used. Based on the values obtained with EBT3 Gafchromic™ film (Figure 4C,D), the valley dose was approximately 2.25–2.5 Gy. The resulting peak-to-valley dose ratio (PVDR) was calculated between 93 and 107. This corresponds well to the PVDR of 85–114 determined based on the CCD camera readouts. The broadbeam dose was between 3.5 and 4.0 Gy, as determined by a soft X-ray chamber.

### 3.7. Visualization of Dose Deposition in the Esophagus after Microbeam Irradiation

Irradiation induces single- or double-strand breaks in the DNA [39]. A suitable marker for DNA damage is the phosphorylation on serin 139 after irradiation of the histone H2AX (γ-H2AX) [40,41]. Previous studies showed that the intensity of the γ-H2AX stain positively correlates with the dose administered and thus is a useful marker for dose deposition in the tissue after MBI [42]. Therefore, γ-H2AX immunostaining was used to detect the radiation-induced DNA damage. Representative fluorescence images are shown in Figure 5A,B Noteworthy, due to the fixation and preparation for immunostaining, the microbeams do not always appear parallel. To visualize the esophageal smooth muscle, we also did a desmin immunostaining (Figure 5C,D).

## 4. Discussion

In conventional thoracic radiotherapy, one of the goals is to minimize the risk of radiogenic damage to the esophagus as an OAR. Depending on the location of the tumor, optimal sparing of the esophagus is not always feasible. In experimental studies, MBI has shown good normal tissue tolerance in the lung concerning thoracic irradiation [31,32]. However, acute toxicity in the esophagus following MBI is unknown so far. To our knowledge, the present study is the first that investigated the radiation effects of MBI on an isolated rat esophagus in an acute ex vivo model. We irradiated isolated esophageal segments and evaluated CCH-induced isometric contractions before and after irradiation. No significant changes regarding baseline tone and maximal contraction force were determined, but the peak latency was found to be increased, and in addition, the maximal force change decreased after MBI. These results align with the findings in cardiac physiological studies [33,34] during and after MBI. In these studies, rodent hearts in the Langendorff perfusion system were irradiated with MBI peak doses up to 400 Gy and 4000 Gy, respectively. Up to MBI peak doses of 400 Gy, no acute or subacute severe effects on cardiac function were observed [33,34]. No significant changes in ventricular or aortic pressure were found, and no structural alterations occurred [34]. Even after irradiation with MBI peak doses of 4000 Gy, only temporary arrhythmia occurred, which converted back to sinus rhythm spontaneously [33]. This may indicate that MBI interfered with the signal transduction cascades of the contraction rather than the contractile apparatus itself, which would agree with the observations made in the current study on the esophagus. Similar physiologic studies were conducted with a rat urinary bladder [43,44,45] or with a human anal sphincter [46] using a conventional linear accelerator. Giglio et al. [43] found that the methacholine and the electrical-field-stimulation (EFS) induced contractions were reduced after irradiation, whereas the contraction in response to potassium chloride (KCl) was not altered. In the study by McDonnell et al. [44], no effects on agonist-induced contractions (CCH and KCl) were found. In mucosal-free bladder strips, EFS-induced contractions were unchanged, and in normal bladder strips, EFS-induced contractions were reduced in a frequency-dependent manner. Similarly, Lorenzi et al. [46] demonstrated an impaired function of the human anal sphincter following radiation therapy for rectal cancer. They found significant differences in response to CCH but not to sodium nitroprusside. In all studies [43,44,46], it was concluded that irradiation affects neuronal structures and intracellular signaling rather than the muscle itself. The same could apply in our preparations since the CCH-induced maximal contraction strength was preserved, but the peak latency and maximal force change were significantly impaired. In contrast, the recent study by Turner et al. [45] reported a KCl-induced decreased contraction of the bladder following irradiation of the rat prostate, and it was speculated that irradiation might also affect the smooth muscle itself. This highlights that the mechanisms of radiogenic injury to muscle remain poorly characterized.

Some limitations of the study should be discussed. First, blood perfusion was acutely interrupted after esophagectomy, resulting in a potential risk of hypoxia. Hypoxic cells were described to be less radiosensitive [47]. In our study, the esophagus was oxygenated by diffusion, which requires a high partial pressure of oxygen. Carbogen (95% O_2_, 5% CO_2_) gassed solution serves this purpose. Good cell survival has been shown in brain tissue [48,49,50], in the Langendorff perfusion system [33,34,51], and in the organ bath [52], especially when using intestinal tissue [53] or esophageal sections [54,55,56]. One study [57] reported toxic effects of 95% O_2_ in prolonged esophageal cell cultures, but these results have not been replicated [58].

Second, the risk of autolytic processes during the esophagus removal from the organ bath for several minutes should be addressed. We immersed the esophagus before and after irradiation into the carbogen-gassed buffer solution at room temperature to keep it as humid as possible. If a marginal degree of autolysis had occurred, the effects of MBI would have been masked by this process rather than exaggerated. In this case, our results would have been underestimated. Furthermore, if significant autolytic processes had occurred, this would have become apparent in both the irradiated and the non-irradiated control tissue (in which the buffer was also removed during sham irradiation). Taken together, neither hypoxia nor autolysis should have significantly confounded our data.

Finally, the short observation time in this study did not allow us to address vascular damage. Radiation-induced vasculopathy is a common late toxicity that can be seen in patients after conventional radiotherapy [59] as well as in mice following MBI [60], leading to fibrosis or necrosis [61]. In organ bath experiments, irradiation did not affect the function of a rabbit aorta [62]. However, inflammation of the endothelium was found 24 h after irradiation [63]. Inflammatory processes may influence the function of the esophageal contractile apparatus. We plan to conduct an in vivo study to investigate the early and late effects of irradiation on the vascular system after conventional and microbeam irradiation.

## 5. Conclusions

In the present study, the function of the contractile apparatus itself was preserved, but the signal transduction was slightly impaired in the middle and distal esophagus segments. Regarding MBI, preserving the cardiac and esophageal function is promising for future therapeutic approaches. From the clinical perspective, our results may plausibly explain radiation-induced motility disorder. Whether this causes symptomatic dysphagia remains to be tested in vivo.

## Figures and Tables

**Figure 1 cells-12-00176-f001:**
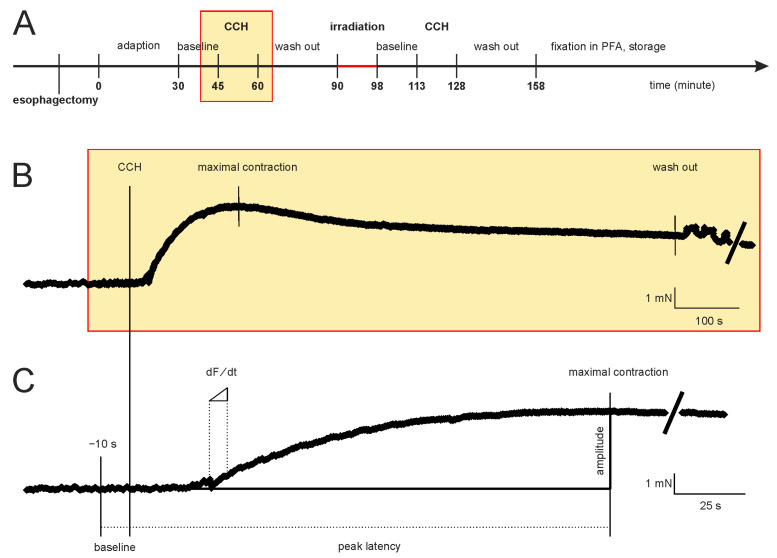
Time course of the experiments and a representative measurement of the contraction force of an esophageal segment. (**A**) Initially, the segments adapted for 30 min to establish stable conditions. Subsequently, the baseline tone, the CCH-induced contraction (yellow rectangle), and the relaxation time (following washout) were registered before and after irradiation. (**B**) Sample recording of the CCH response for 15 min. (**C**) This panel is a section of the force-time diagram of B. Shown is the time interval from the CCH-induced contraction to maximum force. The calculated functional parameters are shown. Abbreviations: CCH = carbachol (adding into the organ bath).

**Figure 2 cells-12-00176-f002:**
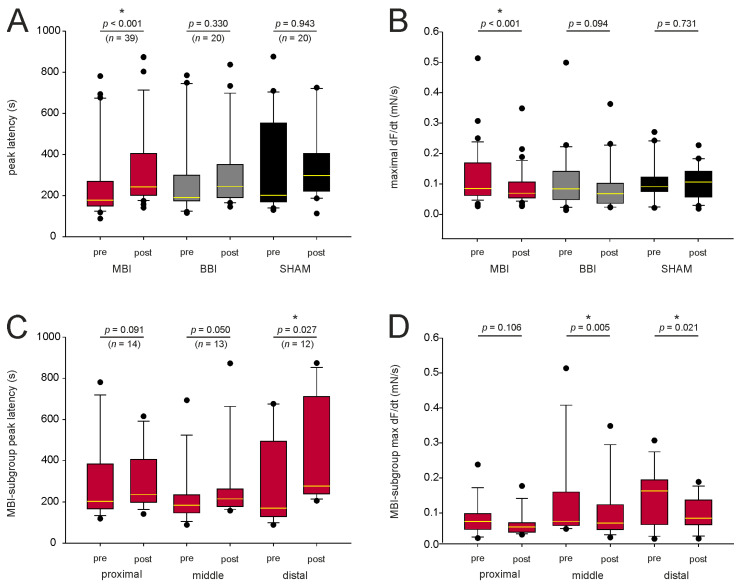
Characterization of the signal transduction. (**A**,**B**) CCH-induced contraction before and after irradiation for mean peak latency and mean force change. In the BBI and SHAM groups, there was no statistical difference (paired *t*-test resp. Wilcoxon test, *p* > 0.05), but mean peak latency significantly increased and mean maximal force change decreased after MBI. (**C**,**D**). In the subgroup analysis, only peak latency for the distal segment (Wilcoxon test, *p* = 0.027) and maximal force change for the middle (Wilcoxon test, *p* = 0.005) and distal (Wilcoxon test, *p* = 0.021) segment remained statistical different. *p*-values were calculated with a paired *t*-test, and, respectively, with a Wilcoxon test. Outliers are plotted as black dots. * *p* < 0.05; The median is illustrated by the yellow horizontal line. Abbreviations: MBI: microbeam irradiation (red boxplot); BBI: broadbeam irradiation (gray boxplot); SHAM: SHAM irradiation (black boxplot); pre-RT and post-RT: before (pre-) and after (post-) irradiation; CCH = carbachol. The number of segments is given in parentheses.

**Figure 3 cells-12-00176-f003:**
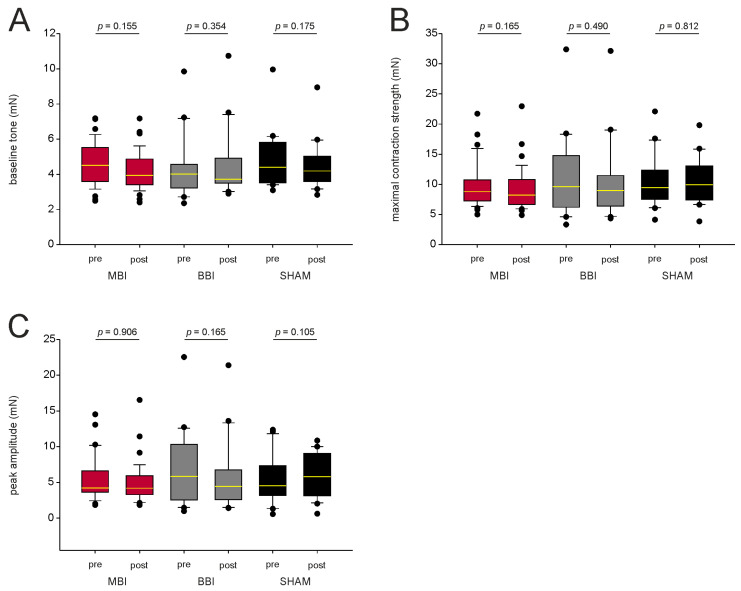
Characterization of the contractile apparatus. (**A**–**C**) CCH-induced contraction before and after irradiation for mean baseline tone, mean maximal contraction strength, and mean peak amplitude. There is no statistical difference before and after irradiation, indicating that MBI and BBI do not affect the contractile apparatus. *p*-values were calculated with a paired *t*-test, and, respectively, with a Wilcoxon test (*p* > 0.05). Outliers are plotted as black dots. The median is illustrated by the yellow horizontal line. Abbreviations: MBI: microbeam irradiation (red boxplot); BBI: broadbeam irradiation (gray box-plot); SHAM: SHAM irradiation (black boxplot); pre-RT and post-RT: before (pre) and after (post) irradiation; CCH = carbachol. The number of segments is given in parentheses.

**Figure 4 cells-12-00176-f004:**
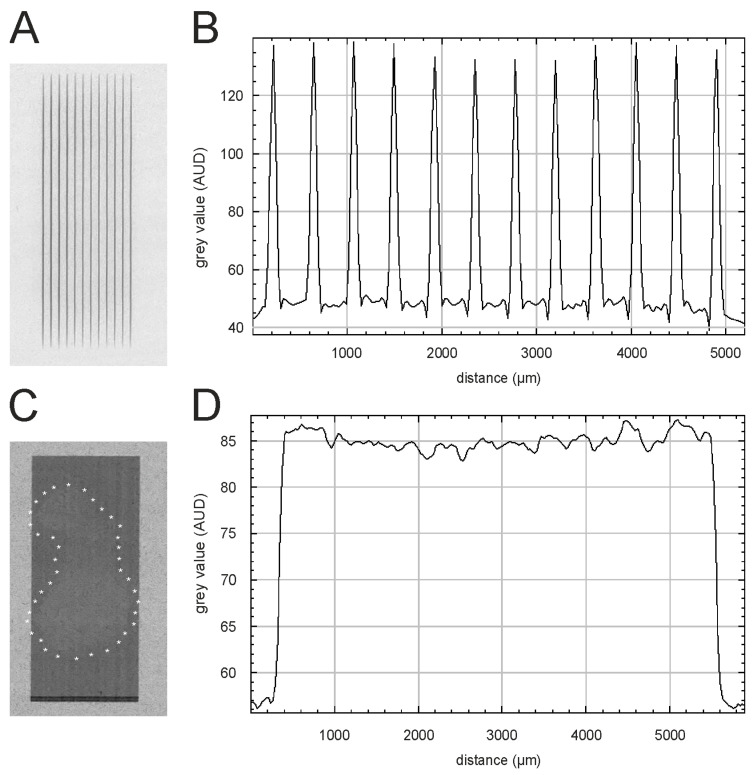
Dosimetry and visualization of the dose deposition in the esophageal segments. (**A**,**B**) Gafchromic™ HD-V2 film recorded during MBI of an esophagus sample and the corresponding beam profile. (**C**,**D**) The outline of the esophagus explant can be seen on the EBT3 Gafchromic™ film after BBI, and the white asterisks outline the sample edges. The corresponding profile of the broadbeam is shown in panel D.

**Figure 5 cells-12-00176-f005:**
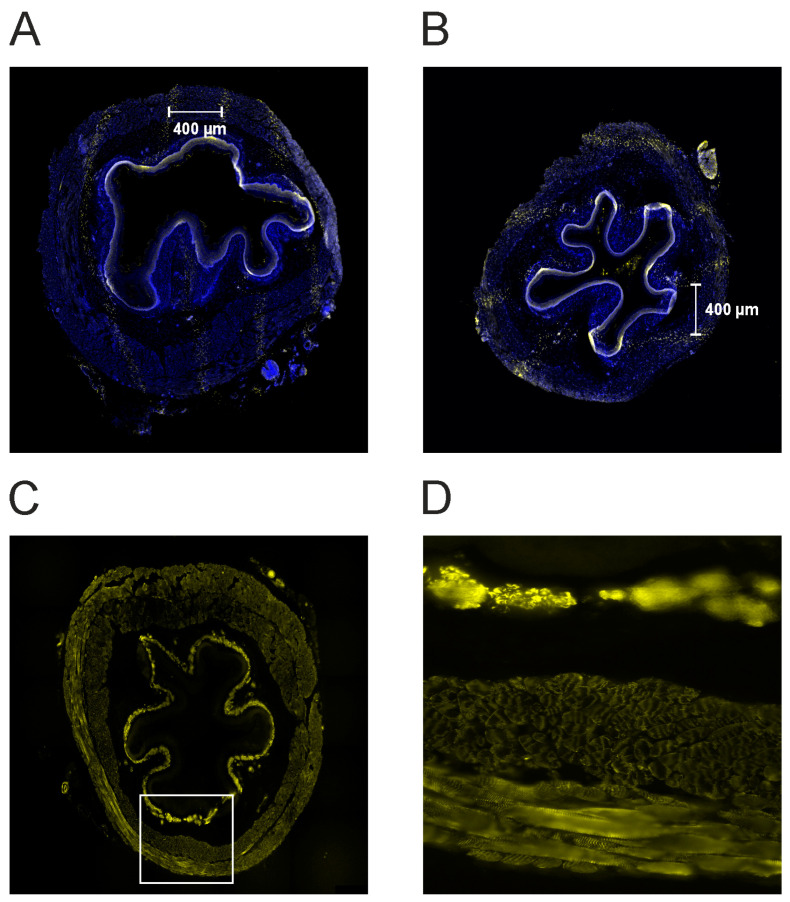
(**A**,**B**) Photographs of the tissue after γH2AX immunostaining for a proximal (panel A) and a middle segment (panel B). Following microbeam irradiation, the immunostaining shows the characteristic pattern of DNA-double strand breaks (yellow dots). This illustrates the dose deposition of the microbeams in the esophagus. The microbeams are separated from each other by approximately 400 µm from center-to-center. (**C**,**D**). Desmin immunostaining for the same segment (see panel A). Panel D is an enlargement of panel C (40-fold, white rectangle in C). The smooth muscle is strongly stained. The different muscle layers can be seen.

**Table 1 cells-12-00176-t001:** Three experimental groups.

Experimental Group	Irradiation	Pharmacological Intervention	Esophageal Segments
I	225 ± 15 Gy	carbachol before and after MBI	proximal (*n* = 14) middle (*n* = 13) distal (*n* = 12)
II	3.5 ± 0.5 Gy	carbachol before and after BBI	proximal (*n* = 6) middle (*n* = 8) distal (*n* = 6)
III	SHAM irradiation	carbachol before and after SHAM	proximal (*n* = 5) middle (*n* = 7) distal (*n* = 8)

MBI = microbeam irradiation; BBI = broadbeam irradiation; SHAM = SHAM irradiation; the number of segments regarding the proximal, middle, and distal segment of the esophagus is given in parentheses.

**Table 2 cells-12-00176-t002:** Functional parameters before irradiation.

Parameter	MBI	BBI	Sham	*p*-Value *
segment length (cm)	0.48 ± 0.02	0.52 ± 0.04	0.53 ± 0.02	0.215
baseline tone (mN)	4.54 ± 0.19	4.35 ± 0.39	4.80 ± 0.35	0.327
maximal contraction strength (mN)	9.91 ± 0.59	10.94 ± 1.49	10.28 ± 0.95	0.939
peak amplitude (mN)	5.37 ± 0.47	6.59 ± 1.18	5.48 ± 0.74	0.983
peak latency (s)	260 ± 30	295 ± 50	343 ± 53	0.305
maximal force change (mN/s)	0.12 ± 0.02	0.11 ± 0.02	0.11 ± 0.02	0.770

MBI = microbeam irradiation; BBI = broadbeam irradiation; SHAM = SHAM irradiation; * Kruskal-Wallis ANOVA on rank test.

## Data Availability

Research data are stored in an institutional repository and will be shared upon request to the corresponding author.

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
