# Peer review of "Effects of Microbeam Irradiation on Rodent Esophageal Smooth Muscle Contraction"

_cells, 2022, doi:10.3390/cells12010176_

Round 1

Reviewer 1 Report

In this manuscript, authors presented well defined and scientifically sound research regarding physiological effects of radiotherapy (microbeam irradiation) in the rodent esophageal smooth muscle contraction. 

The title of the manuscript is concise and relevant. The aim and scope of the study explained well. The abstract is easy to understand and well written. Introduction is quite comprehensive and highlighted work importance as well as its significance towards future prospective. Materials and methods are quite descriptive. Authors provided well explained interpretation of results and discussion. Overall, the article is nicely written with sound performed descriptive experiments.

Before proceeding further, I expect the authors to thoroughly proofread the document and fix all grammatical and typographical errors (some examples include L84, L157, etc)

Minor suggestions:

1)      L66, kindly provides specific peak doses as an example in bracket indicating ‘several hundred’ peak doses.

Author Response

Reviewer 1

In this manuscript, authors presented well defined and scientifically sound research regarding physiological effects of radiotherapy (microbeam irradiation) in the rodent esophageal smooth muscle contraction. 

The title of the manuscript is concise and relevant. The aim and scope of the study explained well. The abstract is easy to understand and well written. Introduction is quite comprehensive and highlighted work importance as well as its significance towards future prospective. Materials and methods are quite descriptive. Authors provided well explained interpretation of results and discussion. Overall, the article is nicely written with sound performed descriptive experiments.

  • We deeply thank the reviewer for carefully reading our manuscript and his comments. We were very pleased about the praise. The manuscript was proofread again by a near-native English speaker, and we now hope that all grammatical errors are corrected.
  • We were asked to remove Figure 2. Now we present the data in a new table (Table 2, line 260).
  • Furthermore, we were able to do a desmin immunostaining, presented in figure 5. We believe this improves our manuscript. Now, the gH2AX immunostaining is shown in figure 5 (line 343).
  • Please note that we have added a new paragraph at the end of the discussion (line 389).

Before proceeding further, I expect the authors to thoroughly proofread the document and fix all grammatical and typographical errors (some examples include L84, L157, etc)

  • We proofread the manuscript once again and we now hope that we corrected all errors (for example: figure 3, contraction strength instead of contraction strenght).

Minor suggestions:

L66, kindly provides specific peak doses as an example in bracket indicating ‘several hundred’ peak doses.

  • We added the information in our manuscript (line 69).

Reviewer 2 Report

Brief summary: aim of the paper, its main contributions and strengths.

This is a study of the functional changes of normal oesophageal smooth muscle tissue after high dose rate radiotherapy. The main goal is an extensive examination, in an ex-vivo model of the rat oesophagus, of the function of the contractile apparatus after experimental radiotherapy, with many controls. The model was exposed to either synchrotron microbeam irradiation (MBI), with peak doses >200 Gy, or to a broad beam (BBI), with a dose between 3.5 to 4 Gy. No severe changes in parameters of oesophageal contraction were determined up to <3 hours after both types of irradiation, with the exception of a delayed function in segments exposed to MBI. The delay is deemed a possible forecast of delayed oesophageal transit after clinical MBI.

General concept comments, concerning areas of weakness, testability of the hypothesis, controls, and other topics.                                         Clinically, a very important “acute” problem (day, weeks) after oesophageal irradiation is dysphagia, due to mucosal damage such as erosions, due to radiation induced cell loss); “delayed” and “late” problems (months, years) generally relate to atrophy of muscle and glands, fibrosis and oesophageal stenosis. Therefore, in the acute phase, morphologic (e.g. histopathologic) spot checks in parallel to the functional investigations would yield an important documentation for the diagnosis and quantification of a concomitant cell loss due to stopped normal perfusion and oxygenation, in other words, to a loss that is likely to relate to oesophageal contraction. The removal of the oesophagus from the organ bath for roughly ten minutes is likely to have accentuated autolysis of cells ex vivo. Further, the degree of oxygenation of the tissues modulates the effects of MBI and BBI. Artificial gassing of oesophageal explants with 95% of oxygen has been shown to be toxic to the tissue, compared to gassing with lower oxygen concentrations (Stoner GD, Pettis W, Haugen A, Jackson F, Harri CC. Explant culture of rat esophagus in a chemically defined medium. In vitro 1981; 17:681-688).

Data of an in vivo study are of interest for the understanding of the radiation damage to smooth muscle elicited in vitro, by means of exposing a hind leg of normal mice to X-ray microbeams similar to those used in the in vitro study, i.e., 50 micron wide beamlets spaced 400 micron on center. Such MBI delivered a peak entrance dose of 312 Gy and a valley minimum dose of 2.7 Gy. Three to six months later, in the path of the beamlets, arterial smooth muscle cells were lost and replaced by fibrous tissue (van der Sanden, Bräuer-Krisch E, Siegbahn EA, Ricard C, Vial JC, Laissue J. Tolerance of arteries to microplanar x-ray beams. Int. J. Radiat Oncol Biol Phys 2010;77:1545–1552). The double strand breaks detected in oesophagus, described in the present paper, document the occurrence of "acute" effects of a comparable MBI. Could the damaged cells be identified as muscle cells in those irradiated oesophageal segments? At what time after irradiation were those double strand breaks shown in Fig. 5 of the present paper identified?

There are differences in the structure of the oesophagus of rats and humans, for instance: Normal oesophageal epithelium in the rat is stratified, squamous, consisting of a basal cell layer, layers of prickle cells, and a thin stratum corneum; there are no submucosal glands. Conversely, the normal human oesophagus has stratified non-keratinized squamous epithelium, as well as submucosal glands. There may also be differences in the configuration and intertwining of the bundles of smooth muscle, and in the presence or absence of striated muscle components. Conceivably, such small variations might influence oesophageal transit times.

Important factors for damage to normal oesophageal tissues, namely vascular parameters such as perfusion and oxygenation of normal oesophageal tissues, are stopped and altered by death, and not really reestablished artificially by a transfer into an organ bath. Both death and culture in vitro are likely to cause some damage to vascular endothelia. Radiation damage will add up to this. For the present study of physiological changes of the contractile apparatus, concomitant vascular changes would deserve particular attention.

Specific comments

Line 25, typo:  …but our results indicate an delayed esophageal function…      

Lines 227 ff. Figure 2 illustrates "the CCH-induced contraction before irradiation". The figure displays six very detailed diagrams, A to F. The corollary is that there was no significant difference between the groups. Might it not suffice to condensate the data in one single short Table for easier reading, without losing critical information?

The same suggestion applies to Figure 4 that illustrates that the alterations of the contractile apparatus by MBI and BBI were not significantly different.

Lines 339 ff. The role of vascular factors (discussed above) deserves mention in the discussion.

Author Response

Reviewer 2

Brief summary: aim of the paper, its main contributions and strengths.

This is a study of the functional changes of normal oesophageal smooth muscle tissue after high dose rate radiotherapy. The main goal is an extensive examination, in an ex-vivo model of the rat oesophagus, of the function of the contractile apparatus after experimental radiotherapy, with many controls. The model was exposed to either synchrotron microbeam irradiation (MBI), with peak doses >200 Gy, or to a broad beam (BBI), with a dose between 3.5 to 4 Gy. No severe changes in parameters of oesophageal contraction were determined up to <3 hours after both types of irradiation, with the exception of a delayed function in segments exposed to MBI. The delay is deemed a possible forecast of delayed oesophageal transit after clinical MBI.

  • We thank the reviewer for the kind and precise summary of our manuscript. The comments were very helpful in improving the manuscript. We have added a new paragraph and a new figure, and now we discuss some limitations.

General concept comments, concerning areas of weakness, testability of the hypothesis, controls, and other topics.                                         

Clinically, a very important “acute” problem (day, weeks) after oesophageal irradiation is dysphagia, due to mucosal damage such as erosions, due to radiation induced cell loss); “delayed” and “late” problems (months, years) generally relate to atrophy of muscle and glands, fibrosis and oesophageal stenosis.

  • In the introduction, we distinguished between acute and late toxicities, but we did not mention fibrosis and stenosis as possible causes of dysphagia. We have now added this to the introduction as suggested by the reviewer (line 45).

Therefore, in the acute phase, morphologic (e.g. histopathologic) spot checks in parallel to the functional investigations would yield an important documentation for the diagnosis and quantification of a concomitant cell loss due to stopped normal perfusion and oxygenation, in other words, to a loss that is likely to relate to oesophageal contraction.

  • The reviewer is right that acute radiation induced esophagitis with typical morphologic changes is one of the most common acute toxicities associated with thoracic radiotherapy. Along this line, hypoxia is probably a major concern that we did not discuss in our manuscript. We added a new paragraph discussing this issue (line 389). Of course, it is right that the perfusion stops after the esophagectomy. Therefore, the tissue has to be oxygenated by diffusion. This requires a relatively high oxygen partial pressure through the specimen, and in almost all organ bath studies, 95% oxygen is used (Wuest 2005, Br J Pharmacol. 2005 Jul;145(5):608-19 ; Mader 2016, Acta Pharmacol Sin . 2016 May;37(5):617-28 ; Ackbar 2012, J Surg Res . 2012 May 1;174(1):56-61 ; Park 2010, Korean J Physiol Pharmacol . 2010 Feb;14(1):29-35).
  • In addition, 95% oxygen was also used in the Langendorff perfusion system employed in a similar experimental setting with microbeam irradiation (Lange 2022, Int J Radiat Oncol Biol Phys . 2022 Sep 1;114(1):143-152 ; Schültke 2022, J Synchrotron Radiat . 2022 Jul 1;29(Pt 4):1027-1032)

The removal of the oesophagus from the organ bath for roughly ten minutes is likely to have accentuated autolysis of cells ex vivo.

  • The reviewer points to the issue of autolysis that may happen following ischemia and/or hypoxia and possibly interacting with radiotolerance. In fact, autolysis is an important aspect in the case of hypoxic cells. In our study, the removal of the esophagus from the organ bath for several minutes might have been subjected to autolysis to some degree. However, we immersed the esophagus before and after irradiation into the carbogen-gassed buffer at room temperature to keep it as humid as possible. We cannot, of course, rule out a marginal degree of autolysis under hypoxia, but even if autolysis had occurred, the effects of MBI would have been masked rather than exaggerated. In this case, our results would have been underestimated rather than artificially produced. Furthermore, this autolysis should have taken place in both irradiated and non-irradiated control tissue (in which the buffer was also removed during sham irradiation). Thus the observed differences between irradiated and non-irradiated tissue were due to irradiation, rather than to autolysis. Taken together, we estimate that it was very unlikely that autolysis could have confounded our data.

Further, the degree of oxygenation of the tissues modulates the effects of MBI and BBI. Artificial gassing of oesophageal explants with 95% of oxygen has been shown to be toxic to the tissue, compared to gassing with lower oxygen concentrations (Stoner GD, Pettis W, Haugen A, Jackson F, Harri CC. Explant culture of rat esophagus in a chemically defined medium. In vitro 1981; 17:681-688).

  • We thank the reviewer for his advice to include this reference (Stoner et al., 1981). In the cited paper, a cytotoxic effect of 95% O2 was detected not before 3 days. In contrast, our experiments were finished after 160 minutes. In another study, however, oesophageal cells in culture (like Stoner 1981) were incubated with 95% oxygen, but no toxic effect was detected (Sohn 2001, Am J Physiol Gastrointest Liver Physiol . 2001 Aug;281(2):G467-78). We feel that this discussion would improve our manuscript, and have therefore added this issue in lines 395.

Data of an in vivo study are of interest for the understanding of the radiation damage to smooth muscle elicited in vitro, by means of exposing a hind leg of normal mice to X-ray microbeams similar to those used in the in vitro study, i.e., 50 micron wide beamlets spaced 400 micron on center. Such MBI delivered a peak entrance dose of 312 Gy and a valley minimum dose of 2.7 Gy. Three to six months later, in the path of the beamlets, arterial smooth muscle cells were lost and replaced by fibrous tissue (van der Sanden, Bräuer-Krisch E, Siegbahn EA, Ricard C, Vial JC, Laissue J. Tolerance of arteries to microplanar x-ray beams. Int. J. Radiat Oncol Biol Phys 2010;77:1545–1552).

  • We thank the reviewer for his suggestion to include this important study, which investigated the late effects of MBI. We included this reference in our discussion (line 412)

The double strand breaks detected in oesophagus, described in the present paper, document the occurrence of "acute" effects of a comparable MBI. Could the damaged cells be identified as muscle cells in those irradiated oesophageal segments? At what time after irradiation were those double strand breaks shown in Fig. 5 of the present paper identified?

  • We deeply regret that we forgot to mention the time of identification of the strand breaks, and we added this information (line 210). The reviewer encouraged us to do a desmin immunostaining. It is shown in figure 5 (line 343).

There are differences in the structure of the oesophagus of rats and humans, for instance: Normal oesophageal epithelium in the rat is stratified, squamous, consisting of a basal cell layer, layers of prickle cells, and a thin stratum corneum; there are no submucosal glands. Conversely, the normal human oesophagus has stratified non-keratinized squamous epithelium, as well as submucosal glands. There may also be differences in the configuration and intertwining of the bundles of smooth muscle, and in the presence or absence of striated muscle components. Conceivably, such small variations might influence oesophageal transit times.

  • It is right that an ex vivo model cannot simply be transferred to humans. In our introduction, studies are cited in which the swallowing function of irradiated patients was examined with scintigraphic or manometric methods. However, the results were not consistent. It would be desirable to study human preparations in an organ bath. From our point of view, it is reasonable to clarify the pathophysiology in ex-vivo and in-vivo experiments first.

Important factors for damage to normal oesophageal tissues, namely vascular parameters such as perfusion and oxygenation of normal oesophageal tissues, are stopped and altered by death, and not really reestablished artificially by a transfer into an organ bath. Both death and culture in vitro are likely to cause some damage to vascular endothelia. Radiation damage will add up to this. For the present study of physiological changes of the contractile apparatus, concomitant vascular changes would deserve particular attention.

  • Some of the factors (perfusion, oxygenation) mentioned by this reviewer, we now discuss in our manuscript. However, we agree with the reviewer that vascular damage in the acute setting may have an influence on the contraction apparatus of esophageal smooth muscle. To this end we are planning an ongoing study, in which we directly analyse irradiation-induced vasculopathy in an “ex ovo” model (e.g. Garcia-Gareta 2020, J Funct Biomater . 2020 Jun 2;11(2):37) as well as in an in-vivo study.

Specific comments

Line 25, typo:  …but our results indicate an delayed esophageal function…     

  • We have carefully proofread our manuscript and hopefully corrected all mistakes.  

Lines 227 ff. Figure 2 illustrates "the CCH-induced contraction before irradiation". The figure displays six very detailed diagrams, A to F. The corollary is that there was no significant difference between the groups. Might it not suffice to condensate the data in one single short Table for easier reading, without losing critical information?

  • That's a good comment. We disscused this issue. Figure 2 has been removed and the data is presented in a table (line 257).

Lines 339 ff. The role of vascular factors (discussed above) deserves mention in the discussion.

  • We have added this issue in our discussion. We thank the reviewer for this important aspect.

Round 2

Reviewer 2 Report

Comments to REVISED MANUSCRIPT and to the ANSWERS TO THE REVIEWER 

- Now, the reviewer acknowledges the following improvements of the manuscript:

- A new paragraph and a new figure have been added to the manuscript, and some limitations to the discussion.

- Fibrosis and stenosis have been mentioned as possible causes of dysphagia (lines 44, 45).

- Figure 2 has been removed and the data are presented in a table (Table 2).

- RE perfusion, hypoxia (lines 419-427):

A new paragraph discussing hypoxia as a major concern has been added to the discussion.

- RE autolysis (428-429):

A sentence about a possible marginal degree of autolysis under hypoxia and its effects has been added to the discussion.

- LAST SENTENCE IN THIS ADDITION: “Taken together, we estimate that it was very unlikely that autolysis could have confounded our data.”

REVIEWER’S COMMENT to the last sentence: Experimental DATA on autolysis – or absence thereof - would be more conclusive than the author’s estimation.

- RE: oxygenation:

It has been stated now that the degree of oxygenation of the tissues may modulate the effects of MBI and BBI (addition in lines 421-427)-

- RE: Vascular damage; relevant statements have been added to the discussion (440 - 447).

- RE: Timing of double strand breaks (four to eight hours after irradiation), illustration of desmin immunoreactivity (lines 364 - 377), and possible destruction of smooth muscle cells by MBI have been added to the manuscript.   

 Typo: Line 426 ….but this hast…    BUT THIS HAS ….

-  In answer to reviewer’s comments: “It is right that an ex vivo model cannot simply be transferred to humans. In our introduction, studies are cited in which the swallowing function of irradiated patients was examined with scintigraphic or manometric methods. However, the results were not consistent. It would be desirable to study human preparations in an organ bath. From our point of view, it is reasonable to clarify the pathophysiology in ex-vivo and in-vivo experiments first.”

QUESTION: How could this study translated in a study of human (esophageal) preparations in an organ bath?

- IN REVISED Manuscript: “Finally, we did not address vascular damage. Radiation-induced vasculopathy is a common late toxicity that can be seen in human after conventional radiotherapy [59] as well as in mice following MBI [60], leading to fibrosis or necrosis [61]. In organ bath experiments, irradiation did not affect the function of a rabbit aorta [62]. However, vasculopathy may further influence the contractile esophageal apparatus. We are planning an ongoing study in which we investigate early and late effects of irradiation on the vascular 446 system after conventional and microbeam irradiation.” (Revised manuscript, lines 440 – 447)

 COMMENT: A longer-term in vivo study might indeed answer many questions raised in this study and its review.